# Chromatin accessibility dynamics across *C. elegans* development and ageing

Jürgen Jänes[1,2†], Yan Dong[1,2†], Michael Schoof[1,2‡], Jacques Serizay[1,2‡], Alex Appert[1,2], Chiara Cerrato[1,2], Carson Woodbury[1,2], Ron Chen[1,2§], Carolina Gemma[1,2#], Ni Huang[1,2], Djem Kissiov[1,2¶], Przemyslaw Stempor[1,2], Annette Steward[1,2], Eva Zeiser[1,2], Sascha Sauer[3,4], Julie Ahringer[1,2*]

[1]Department of Genetics, University of Cambridge, Cambridge, United Kingdom; [2]The Gurdon Institute, University of Cambridge, Cambridge, United Kingdom; [3]Max Delbrück Center for Molecular Medicine, Berlin, Germany; [4]Max Planck Institute for Molecular Genetics, Otto-Warburg Laboratories, Berlin, Germany

**\*For correspondence:**
ja219@cam.ac.uk

[†]These authors contributed equally to this work
[‡]These authors also contributed equally to this work

**Present address:** [§]School of Molecular and Cellular Biology, Faculty of BiologicalSciences, University of Leeds, Leeds, United Kingdom; [#]Department of Surgery and Cancer, Imperial College London, London, United Kingdom; [¶]University of California, Berkeley, United States

**Abstract** An essential step for understanding the transcriptional circuits that control development and physiology is the global identification and characterization of regulatory elements. Here, we present the first map of regulatory elements across the development and ageing of an animal, identifying 42,245 elements accessible in at least one *Caenorhabditis elegans* stage. Based on nuclear transcription profiles, we define 15,714 protein-coding promoters and 19,231 putative enhancers, and find that both types of element can drive orientation-independent transcription. Additionally, more than 1000 promoters produce transcripts antisense to protein coding genes, suggesting involvement in a widespread regulatory mechanism. We find that the accessibility of most elements changes during development and/or ageing and that patterns of accessibility change are linked to specific developmental or physiological processes. The map and characterization of regulatory elements across *C. elegans* life provides a platform for understanding how transcription controls development and ageing.
DOI: https://doi.org/10.7554/eLife.37344.001

## Introduction

The genome encodes the information for organismal life. Because the deployment of genomic information depends in large part on regulatory elements such as promoters and enhancers, their identification and characterization is essential for understanding genome function and its regulation.

Regulatory elements are typically depleted for nucleosomes, which facilitates their identification using sensitivity to digestion by nucleases such as DNase I or Tn5 transposase, termed DNA accessibility (*Sabo et al., 2006*; *Crawford et al., 2006*; *Buenrostro et al., 2013*). In different organisms, large repertoires of regulatory elements have been determined by profiling DNA accessibility genome-wide in different cell types and developmental stages (*Thomas et al., 2011*; *Kharchenko et al., 2011*; *Thurman et al., 2012*; *Yue et al., 2014*; *Kundaje et al., 2015*; *Daugherty et al., 2017*; *Ho et al., 2017*). However, no study has yet investigated regulatory element usage across the life of an animal, from the embryo to the end of life. Such information is important, because different transcriptional programs operate in different periods of life and ageing. *Caenorhabditis elegans* is ideal for addressing this question, as it has a simple anatomy, well-defined cell types, and short development and lifespan. A map of regulatory elements and their temporal dynamics would facilitate understanding of the genetic control of organismal life.

Active regulatory elements have previously been shown to have different transcriptional outputs and chromatin modifications (*Andersson, 2015*; *Kim and Shiekhattar, 2015*). Transcription is initiated at both promoters and enhancers, with most elements having divergent initiation events from

two independent sites (*Core et al., 2008*; *Kim et al., 2010*; *De Santa et al., 2010*; *Koch et al., 2011*; *Chen et al., 2013*). However, promoters and enhancers differ in the production of stable transcripts. At protein-coding promoters, productive transcription elongation produces a stable transcript, whereas enhancers and the upstream divergent initiation from promoters generally produce short, aborted, unstable transcripts (*Core et al., 2014*; *Andersson et al., 2014*; *Rennie et al., 2017*).

Promoters and enhancers have also been shown to be differently enriched for specific patterns of histone modifications. In particular, promoters often have high levels of H3K4me3 and low levels of H3K4me1, whereas enhancers tend to have the opposite pattern of higher H3K4me1 and lower H3K4me3 (*Heintzman et al., 2007*; *Heintzman et al., 2009*). However, in human and *Drosophila* cell lines, it was observed that H3K4me3 and H3K4me1 levels correlate with levels of transcription at regulatory elements, rather than whether the element is a promoter or an enhancer (*Core et al., 2014*; *Henriques et al., 2018*; *Rennie et al., 2018*). Further, analyses of genes that are highly regulated in development showed that their promoters lacked chromatin marks associated with activity (including H3K4me3), even when the associated genes are actively transcribed (*Zhang et al., 2014*; *Pérez-Lluch et al., 2015*). Therefore, stable elongating transcription, rather than histone modification patterns, appears to be the defining feature that distinguishes active promoters from active enhancers (reviewed in *Andersson, 2015*; *Andersson et al., 2015*; *Kim and Shiekhattar, 2015*; *Henriques et al., 2018*; *Rennie et al., 2018*).

Regulatory elements have not been systematically mapped and annotated in *C. elegans*. Promoter identification has been hampered because the 5' ends of ~70% of protein-coding transcripts are trans-spliced to a 22nt leader sequence (*Allen et al., 2011*). Because the region from the transcription initiation site to the trans-splice site (the 'outron') is removed and degraded, the 5' end of the mature mRNA does not mark the transcription start site. To overcome this difficulty, previous studies identified transcription start sites for some genes through profiling transcription initiation and elongation in nuclear RNA or by inhibiting *trans*-splicing at a subset of stages (*Gu et al., 2012*; *Chen et al., 2013*; *Kruesi et al., 2013*; *Saito et al., 2013*). In addition, two recent studies used ATAC-seq or DNAse I hypersensitivity to map regions of accessible chromatin in some developmental stages, and predicted element function by proximity to first exons or chromatin state (*Daugherty et al., 2017*; *Ho et al., 2017*).

Toward building a comprehensive map of regulatory elements and their use during the life of an animal, here we used multiple assays to systematically identify and annotate accessible chromatin in the six *C. elegans* developmental stages and at five time points of adult ageing. Strikingly, most elements undergo a significant change in accessibility during development and/or ageing. Clustering the patterns of accessibility changes in promoters reveals groups that act in shared processes. This map makes a major step toward defining regulatory element use during *C. elegans* life.

## Results and discussion

### Defining and annotating regions of accessible DNA

To define and characterize regulatory elements across *C. elegans* life, we collected biological replicate samples from a developmental time course and an ageing time course (*Figure 1A*). The developmental time course consisted of wild-type samples from six developmental stages (embryos, four larval stages, and young adults). For the ageing time course, we used *glp-1(e2144ts)* mutants to prevent progeny production, since they lack germ cells at the restrictive temperature. Five adult ageing time points were collected, starting from the young adult stage (day 1) and ending at day 13, just before the major wave of death.

*Figure 1A* outlines the datasets generated. For all developmental and ageing time points, we used ATAC-seq to identify accessible regions of DNA. We also sequenced strand-specific nuclear RNA (>200 nt long) to determine regions of transcriptional elongation, because previous work demonstrated that this approach could capture outron signal linking promoters to annotated exons (*Chen et al., 2013*; *Kruesi et al., 2013*; *Saito et al., 2013*). For the development time course, we additionally sequenced short (<100 nt) capped nuclear RNA to profile transcription initiation, profiled four histone modifications to characterize chromatin state (H3K4me3, H3K4me1, H3K36me3, and H3K27me3), and performed a DNase I concentration course to investigate the relative

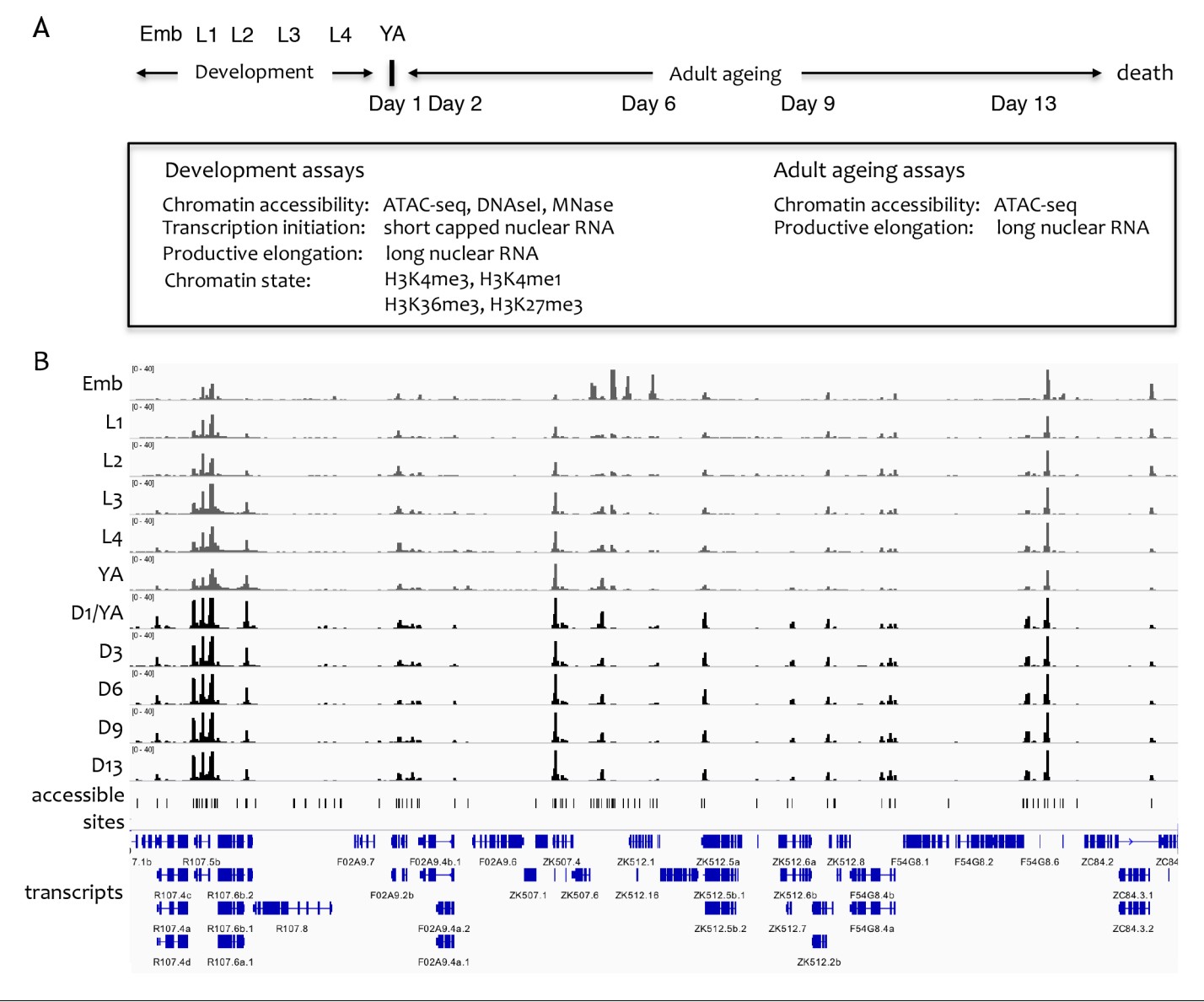

**Figure 1.** Overview of the project. (**A**) Overview of genome-wide assays and time points of developmental and ageing samples. For development samples, chromatin accessibility, transcription initiation, productive elongation, and chromatin state were profiled in six stages of wild-type animals (embryos, four larval stages, young adults). For ageing samples, chromatin accessibility and productive transcription elongation were profiled in five time points of sterile adult *glp-1* mutants (Day 1/Young adult, Day 2, Day 6, Day 9, Day 13). (**B**) Representative screen shot of normalized genome-wide accessibility profiles in the eleven samples (chrIII:9,041,700–9,196,700, 154 kb).

DOI: https://doi.org/10.7554/eLife.37344.002

The following source data and figure supplements are available for figure 1:

**Source data 1.** Accessible sites identified using ATAC-seq.
DOI: https://doi.org/10.7554/eLife.37344.006

**Figure supplement 1.** Comparison of ATAC-seq to concentration courses of DNase I-seq and MNase-seq.
DOI: https://doi.org/10.7554/eLife.37344.003

**Figure supplement 2.** Reproducibility and broad relatedness of ATAC-seq and RNA-seq data.
DOI: https://doi.org/10.7554/eLife.37344.004

**Figure supplement 3.** Reproducibility and broad relatedness of the histone modification data.
DOI: https://doi.org/10.7554/eLife.37344.005

accessibility of elements. Micrococcal nuclease (MNase) data were also collected for the embryo stage. As previously noted by others, we found that ATAC-seq accessibility signal is similar to that observed using a low-concentration DNase I or MNase, and that the ATAC-seq data has the highest signal-to-noise ratio (*Buenrostro et al., 2013*); *Figure 1—figure supplement 1C*) (*Buenrostro et al., 2013*; *Figure 1—figure supplement 1A*).

To define sites that are accessible in at least one developmental or ageing stage, focal peaks of significant ATAC-seq enrichment were identified across all developmental and ageing samples, yielding 42,245 individual elements (*Figure 1B*, *Figure 1—source data 1*; see Materials and methods for details). Of these, 72.8% overlap a transcription factor binding site (TFBS) mapped by the modENCODE or modERN projects (*Araya et al., 2014*; *Kudron et al., 2018*), supporting their potential regulatory functions (*Figure 2—figure supplement 1A*).

Two recent studies reported accessible regions in *C. elegans* identified using DNase I hypersensitivity or ATAC-seq (*Ho et al., 2017*; *Daugherty et al., 2017*). The 42,245 accessible elements defined here overlap 33.7% of (*Ho et al., 2017*) DNase I hypersensitive sites and 47.9% of (*Daugherty et al., 2017*) ATAC-seq peaks (*Figure 2—figure supplement 1B,C*). Examining the non-overlapping sites from pairwise comparisons, it appears that differences in peak calling methods account for some of the differences. Accessible regions determined here required a focal peak of enrichment, whereas the other studies found both focal sites and broad regions with increased signal. Consistent with these differences in methods, sites unique to the two studies are enriched for exonic chromatin, depleted for both TFBS and transcription initiation sites, and often found in broad regions of increased accessibility across transcriptionally active gene bodies (*Figure 2—figure supplement 1B–E*). Similarly, using MACS2 to call peaks on the ATAC-seq data reported here, as used by *Daugherty et al. (2017)*, identified a group of exon enriched sites not found using our peak calling method (*Figure 2—figure supplement 2A*). However, the fraction of such sites is relatively small indicating that other differences also contribute, such as signal-to-noise or nematode growth methods.

To functionally classify elements, we annotated each of the 42,245 elements for transcription initiation and transcription elongation signals on both strands (*Figure 2A,B*; *Figure 2—source data 1*; see Materials and methods for details). Overall, 37.1% of elements had promoter activity, defined by a significant increase in transcription elongation signal originating at the element in at least one stage and one direction. Promoters were assigned to protein-coding or pseudogenes if continuous transcription elongation signal extended from the element to an annotated first exon (covering the outron). Promoters were unassigned if transcription elongation signal was not linked to an annotated gene. We observed detectable transcription initiation signal at 82.3% of elements (*Figure 2—source data 1*); those with no significant transcription elongation signal in either direction were annotated as putative enhancers (hereafter referred to as 'enhancers'). The remaining elements had no detectable transcriptional activity or overlapped ncRNAs (tRNA, snRNA, snoRNA, rRNA, or miRNA) (*Figure 2B*; *Figure 2—source data 1*). We found that accessible sites are enriched for being located within outrons or intergenic regions (*Figure 2—figure supplement 3*).

Within the promoter class, we defined 15,572 protein-coding coding promoters: 11,478 elements are unidirectional promoters and 2118 are divergent promoters that drive expression of two oppositely oriented protein-coding genes (*Figure 2—source data 1*). In total, promoters were defined for 11,196 protein-coding genes, with 3000 genes having >1 promoter (*Figure 2C*). The protein-coding promoter annotations show good overlap with four sets of TSSs previously defined based on mapping transcription (*Chen et al., 2013*; *Kruesi et al., 2013*; *Saito et al., 2013*; *Gu et al., 2012*); 76.8–85.1%; *Figure 2—figure supplement 5*). Enhancers (n = 19,231) were assigned to a gene if they are located within the region from its most upstream promoter to its gene end; 6668 genes have at least one associated enhancer, and 3240 genes have >1 enhancer (*Figure 2C*).

The locations of unassigned promoters (n = 3106) suggest different potential functions. A large fraction (35.1%) generate antisense transcripts within the body of a protein coding gene, suggesting a possible role in regulating expression of the associated gene (*Figure 2—figure supplement 5*). Another large group (38.4%) produce antisense transcripts from an element that is a protein coding promoter in the sense direction, a pattern seen in many mammalian promoters, termed upstream antisense (uaRNA) or promoter upstream (PROMPT) transcripts (*Figure 2—figure supplement 5*; *Preker et al., 2008*; *Flynn et al., 2011*; *Sigova et al., 2013*). Most of the rest (21.7%) are intergenic and may define promoters for unannotated transcripts.

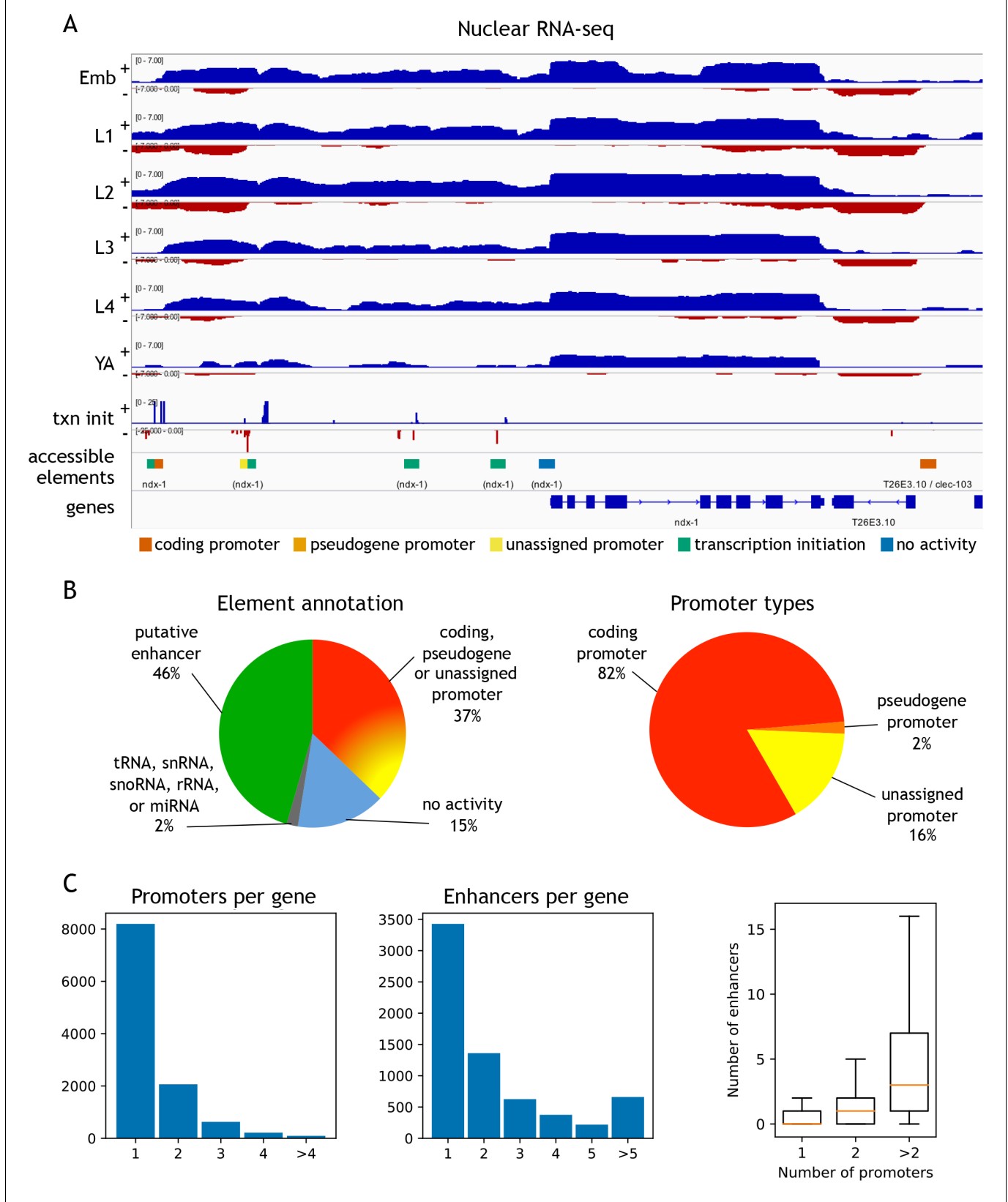

**Figure 2.** Annotation of accessible elements. (**A**) Top, strand-specific nuclear RNA in each developmental stage monitors transcription elongation; plus strand, blue; minus strand, red. Below is transcription initiation signal, accessible elements (colored by annotation), and gene models (chrI:12,675,000–

*Figure 2 continued on next page*

*Figure 2 continued*

12,683,400, 8.4 kb). The left side of each element is colored by the reverse strand annotation whereas the right side of an element is colored by the forward strand annotation (color key at bottom). (B) Left, distribution of accessible sites in four categories: promoters (one or both strands), putative enhancers, no activity, or overlapping a tRNA, snRNA, snoRNA, rRNA, or miRNA. Right, distribution of different types of promoter annotations. (C) Left, distribution of the number of promoters and enhancers per gene; right, boxplot shows that genes with more promoters also have more enhancers.

DOI: https://doi.org/10.7554/eLife.37344.007

The following source data and figure supplements are available for figure 2:

**Source data 1.** Regulatory annotation of accessible sites.
DOI: https://doi.org/10.7554/eLife.37344.014

**Figure supplement 1.** Comparisons to previous accessibility maps.
DOI: https://doi.org/10.7554/eLife.37344.008

**Figure supplement 2.** Effect of differences in peak calling methods on the types of identified accessible sites.
DOI: https://doi.org/10.7554/eLife.37344.009

**Figure supplement 3.** Genomic locations of accessible sites.
DOI: https://doi.org/10.7554/eLife.37344.010

**Figure supplement 4.** Comparison to published TSS maps.
DOI: https://doi.org/10.7554/eLife.37344.011

**Figure supplement 5.** Types of unassigned promoters.
DOI: https://doi.org/10.7554/eLife.37344.012

**Figure supplement 6.** Transgenic tests of annotated promoters and enhancers for promoter activity.
DOI: https://doi.org/10.7554/eLife.37344.013

## Patterns of histone marks at promoters and enhancers

Promoters and enhancers show general differences in patterns of histone modifications, such as higher levels of H3K4me3 at promoters or H3K4me1 at enhancers, and chromatin states are frequently used to define elements as promoters or enhancers (*Heintzman et al., 2007*; *Ernst and Kellis, 2010*; *Ernst et al., 2011*; *Kharchenko et al., 2011*; *Hoffman et al., 2013*; *Daugherty et al., 2017*). However, it has been shown that H3K4me3 levels correlate with transcriptional activity rather than with function (*Pekowska et al., 2011*; *Core et al., 2014*; *Andersson et al., 2014*; *Henriques et al., 2018*; *Rennie et al., 2018*), suggesting that defining regulatory elements solely based on chromatin state is likely to lead to incorrect annotations.

To further investigate the relationship between chromatin marking and element function, we mapped four histone modifications at each developmental stage (H3K4me3, H3K4me1, H3K27me3, H3K36me3) and examined their patterns around coding promoters and enhancers. As expected, many coding promoters had high levels of H3K4me3 and were depleted for H3K4me1 (*Figure 3A*). Moreover, enhancers had generally low levels of H3K4me3 and higher levels of H3K4me1 than promoters (*Figure 3A*). However, many elements did not have these patterns. For example, about 50% of coding promoters have a high level of H3K4me1 and no or low H3K4me3 marking (*Figure 3A*).

To investigate the nature of these patterns, we examined coefficients of variation of gene expression (CV; *Gerstein et al., 2014*) of the associated genes. Genes with broad stable expression across cell types and development, such as housekeeping genes, have low variation of gene expression levels and hence a low CV value. In contrast, genes with regulated expression, such as those expressed only in particular stages or cell types have a high CV value. We found a strong inverse correlation between a gene's CV value and its promoter H3K4me3 level ($-0.64$, $p < 10^{-15}$, Spearman's rank correlation; *Figure 3*; *Figure 3—figure supplement 1A*). Furthermore, promoters with low or no H3K4me3 marking are enriched for H3K27me3 (*Figure 3*; *Figure 3—figure supplement 1A*), which is associated with regulated gene expression (*Tittel-Elmer et al., 2010*; *Pérez-Lluch et al., 2015*; *Evans et al., 2016*). These results support the view that H3K4me3 marking may be a specific feature of promoters with broad stable activity, consistent with the finding that active promoters of regulated genes lack H3K4me3 (*Pérez-Lluch et al., 2015*). The profiling here was done in whole animals, which may have precluded detecting modifications occurring in a small number of nuclei. Nevertheless, the results indicate that chromatin state alone is not a reliable metric for element annotation. Histone modification patterns at many promoters resemble those at enhancers, and vice versa.

Promoters and enhancers also share sequence features. Both are enriched for initiator INR elements, although enhancers have a slightly lower INR frequency (*Figure 3B* and *Figure 3—figure*

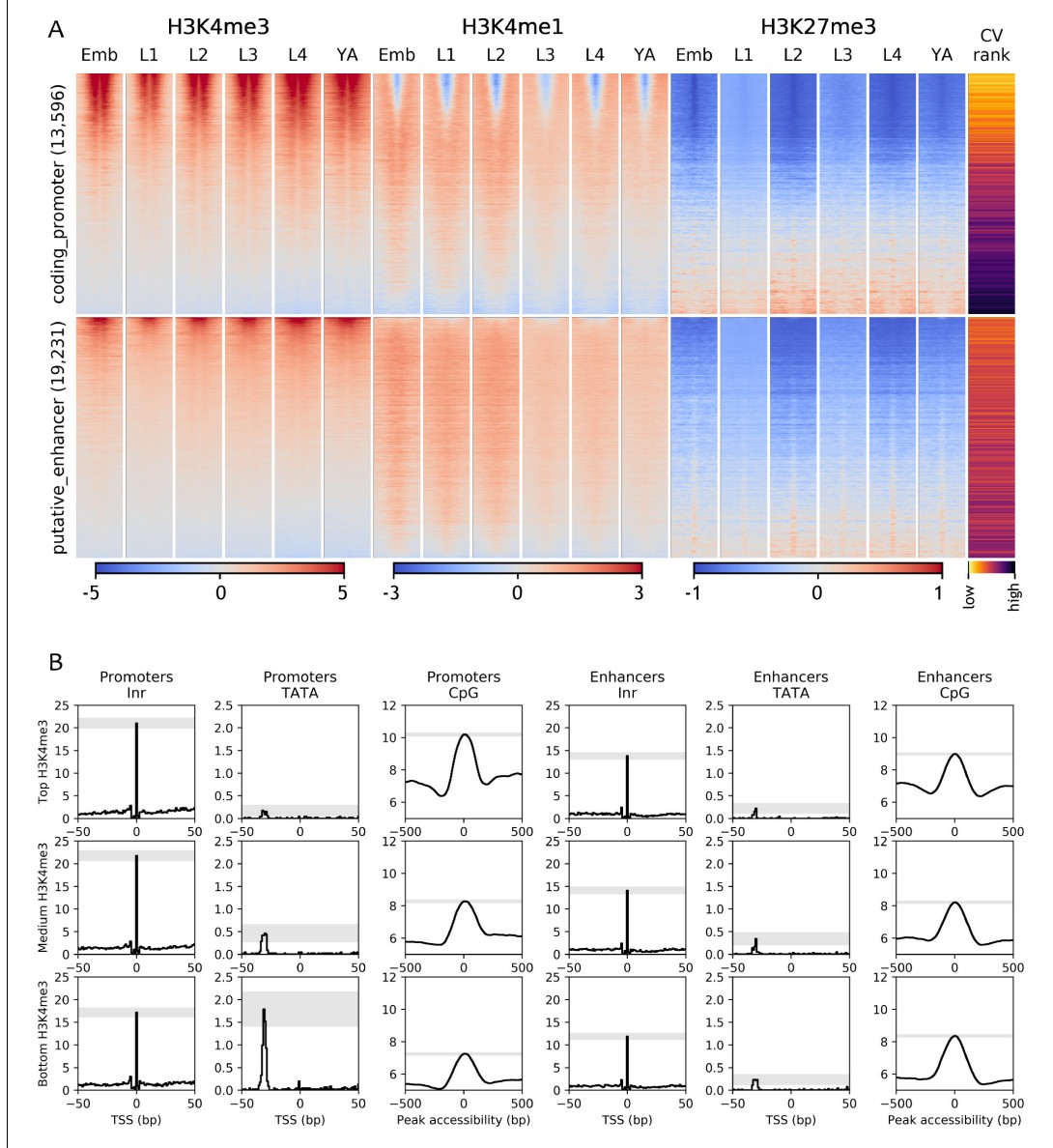

**Figure 3.** Chromatin state and sequence features of promoters and enhancers. (**A**) Heatmaps of indicated histone modifications and CV values at coding promoters (top), and enhancers (bottom), aligned at element midpoints. Elements are ranked by mean H3K4me3 levels. Low CV values indicate broad expression across development and cell types and high CV values indicate regulated expression. Promoters of genes with low CV values have high H3K4me3 levels. (**B**) Distribution of initiator Inr motif, TATA motif, and CpG content at coding promoters and enhancers, separated by H3K4me3 level (top, middle, and bottom thirds). Grey-shaded regions represent 95% confidence intervals of the sample mean at the genomic position with the highest signal.

DOI: https://doi.org/10.7554/eLife.37344.015

The following figure supplement is available for figure 3:

**Figure supplement 1.** Chromatin state and sequence features of promoters and enhancers sorted by CV value.

DOI: https://doi.org/10.7554/eLife.37344.016

*supplement 1B*). Promoters and enhancers are also both enriched for CpG dinucleotides (*Figure 3B* and *Figure 3—figure supplement 1B*). Promoters with high H3K4me3 and low CV values (broadly expressed genes) have the highest CpG content, whereas those with low H3K4me3 and high CV values have the lowest CpG content (*Figure 3B* and *Figure 3—figure supplement 1B*). Promoters also differ from enhancers by the presence of TATA motifs, which occur predominantly at genes with low

H3K4me3,and high CV values (i.e. with regulated expression; *Figure 3B* and *Figure 3—figure supplement 1B*).

## Promoters and enhancers can drive gene expression in an orientation independent manner

To validate the promoter annotations, we compared them with studies where small regions of DNA had been defined as promoters using transgenic assays. These comprised 10 regions are defined based on transcription initiation signal (*Chen et al., 2014*), nine regions defined based on proximity to a germ line gene (*Merritt et al., 2008*), and four defined by proximity to the first exon of a muscle expressed gene (*Hunt-Newbury et al., 2007*). Of these 23 regions, 21 overlap an element in our set of accessible sites, 19 of which are annotated as protein coding promoters (*Figure 2—figure supplement 6A*). One of the remaining two is annotated as an enhancer and the other overlaps an accessible element for which no transcriptional signal was detected. We further directly tested three elements annotated as promoters (for *hlh-2*, *ztf-11* and *bed-3* genes), and found that all three drove robust expression of a histone-GFP reporter (*Figure 2—figure supplement 6A*). Overall, there is good concordance between promoter annotation and promoter activity.

Most of the elements annotated as protein-coding promoters are flanked by bidirectional transcription initiation signal (74.0%), similar to the pattern seen in mammals. Most (82.6%) are unidirectional promoters, producing a protein-coding transcript in one direction, but no stable transcript from the upstream initiation site. To test whether such upstream antisense initiation sites could function as promoters, we inverted the orientation of two active unidirectional promoters (*ztf-11* and F58D5.5). If the lack of in vivo transcription elongation was a property of the element or initiation site itself, the GFP fusion should not be expressed. However, we observed that the two inverted unidirectional promoters both drove GFP expression. The expression patterns generated were similar in both orientations, although the *ztf-11* promoter was weaker when inverted (*Figure 2—figure supplement 6B,C*). These results suggest that signals for productive elongation occur downstream of the transcription initiation site.

Similar to the upstream antisense transcription initiation observed at promoters, enhancers also show transcription initiation signals but generally do not produce stable transcripts (*Core et al., 2014*; *Andersson et al., 2014*). Previous studies have reported that some enhancers can function as promoters in transgenic assays and also at endogenous loci (*Kowalczyk et al., 2012*; *Leung et al., 2015*; *Nguyen et al., 2016*; *van Arensbergen et al., 2017*; *Mikhaylichenko et al., 2018*). To assess the potential promoter activities of *C. elegans* enhancers, we directly fused 12 putative enhancers that had transcription initiation signal in embryos to a histone-GFP reporter gene and assessed transgenic strains for embryo expression. Two of the tested enhancers are located in introns, and one of these, from the *bro-1* gene, has been previously validated as an enhancer (*Brabin et al., 2011*); most of the others are associated with the *hlh-2* or *ztf-11* genes. We found that 10 of 12 tested regions drove reporter expression in embryos, including the two intronic enhancers (*Figure 2—figure supplement 6B,C*). Whereas the *hlh-2* and *ztf-11* promoters drove strong, broad expression, the associated enhancers were active in a smaller number of cells and expression levels were overall lower (*Figure 2—figure supplement 6B,C*). We also tested two enhancers in inverted orientation and found that both showed similar activity in both orientations, as observed for the two tested promoters (*Figure 2—figure supplement 6B,C*). The percentage of enhancers that functioned as active promoters is higher than that observed in a cell-based assay (*Nguyen et al., 2016*), possibly because all cell types are tested in an intact animal. Episomal-based assays have also been reported to underestimate activity (*Inoue et al., 2017*).

## Extensive regulation of chromatin accessibility in development

We observed extensive changes in chromatin accessibility across development, with most elements showing a significant difference within the developmental time course (71%,>=2 fold change, FDR < 0.01; *Figure 4—source data 1*; see Materials and methods). To investigate how accessibility relates to gene expression, we focused on the 13,596 elements annotated as protein-coding promoters. Of these, 10,199 displayed significant changes in accessibility in development, with the remaining 3397 promoters classified as having stable accessibility. We note that the detected changes could be due to regulation of accessibility, or alternatively to changes in cell number during

development (e.g. the number of germ line nuclei increases from two in L1 larvae to ~2000 in young adults).

We reasoned that promoters having similar patterns of accessibility changes over development may regulate genes that function in shared processes and be regulated by shared sets of transcription factors. To investigate this, we applied *k*-medoid clustering to the 10,199 promoters with developmental changes in accessibility, defining 16 clusters (*Figure 4A*, *Figure 4—figure supplement 1*, *Figure 4—figure supplement 2*, and *Figure 4—source data 1*; see Materials and methods). Within clusters, we observed that promoter accessibility and nuclear RNA levels are usually correlated (mean r = 0.47 (sd = 0.11) across all clusters), indicating that accessibility is a good metric of promoter activity and overall gene expression (*Figure 4—figure supplement 1* and *Figure 4—figure supplement 2*).

To investigate whether the shared patterns of accessibility changes over development identify promoters of genes involved in common processes, we took advantage of recent single-cell profiling data obtained from L2 larvae, which provides gene expression measurements in different tissues (*Cao et al., 2017*). We find that half of the developmental promoter clusters are enriched for genes with tissue biased expression (*Figure 4A*, *Figure 4—figure supplement 1* and *Figure 4—figure supplement 2*). Based on these patterns of enrichment, we defined four gonad promoter clusters (G1-G4), two intestine clusters (I1, I2), one hypodermal cluster (H) and one cluster enriched for neural and muscle expression (N + M) (*Figure 4A*, *Figure 4—figure supplement 1* and *Figure 4—figure supplement 2*). Genes associated with the remaining eight promoter clusters (Mix1–8) are generally expressed in multiple tissues, but predominantly in the soma (*Figure 4A*, *Figure 4—figure supplement 1* and *Figure 4—figure supplement 2*). As expected, genes linked to the stable promoters are widely expressed. Interestingly, within a tissue, promoter clusters can exhibit similar variations in accessibility but with different amplitude. For instance, gonad clusters G1 and G2 both show a sharp increase in accessibility at the L3 stage; however, the increase is 1.5-fold larger in G2 than in G1. The gonad clusters are generally characterized by an increase of promoter accessibility starting in L3 when germ cell number strongly increases.

To further investigate promoter clusters sharing accessibility dynamics, we performed Gene Ontology analyses on the associated genes. As expected, we found that clusters containing genes enriched for expression in a particular tissue are also associated with GO terms related to that tissue (*Figure 4A*, *Figure 4—figure supplement 1* and *Figure 4—figure supplement 2*). For instance, cluster H contains genes highly expressed in hypodermis and GO terms linked to cuticle development. Of note, the four accessibility clusters enriched for expression in germ line are associated with GO terms for different sets of germ line functions (*Figure 4—figure supplement 1* and *Figure 4—figure supplement 2*). Similarly, the two intestinal clusters also identify genes with different types of intestinal function. Furthermore, accessibility dynamics can reflect the temporal function of the associated promoters. For instance, cluster Mix4 has GO terms indicative of neuronal development and highest accessibility in the embryo, when many neurons develop. These results suggest that promoter clusters contain genes acting in a shared process and having a similar mode of regulation.

To identify potential transcriptional regulators, we asked whether the binding of particular transcription factors is enriched in any promoter clusters, using TF binding data from the modENCODE and modERN projects (*Boyle et al., 2014*; *Kudron et al., 2018*). TFs with enriched binding were found for each cluster (*Figure 5A*), and the expression of such TFs was generally enriched in the expected tissue. For example, we found that ELT-2, an intestine-specific GATA protein (*Fukushige et al., 1998*), has enriched binding at promoters in intestinal clusters 1 and 2. Similarly, hypodermal transcription factors BLMP-1 (*Horn et al., 2014*), NHR-25 (*Gissendanner and Sluder, 2000*) and ELT-3 (*Gilleard et al., 1999*) are enriched in the hypodermal promoter cluster, and binding of the germ line XND-1 factor (*Wagner et al., 2010*) is enriched in the germ line clusters of promoters. We also identified novel tissue-specific associations for uncharacterized transcription factors, such as ZTF-18 and ATHP-1 with germ line promoter clusters and CRH-2 with the intestinal clusters (*Figure 5A*). These results agree and extend those of *Cao et al. (2017)*, who identified TFs for which binding was correlated with cell-type-specific expression levels.

We also observed differences in TF-binding enrichments between promoter clusters associated with the same tissue. For example, Clusters G1-G4 all contain promoters associated with germline-enriched genes (*Figure 4A*). However, distinct binding enrichments are observed in promoters in G1-G2 compared to those in G3-G4, with the latter showing enrichment for LIN-35 and DPL-1, two

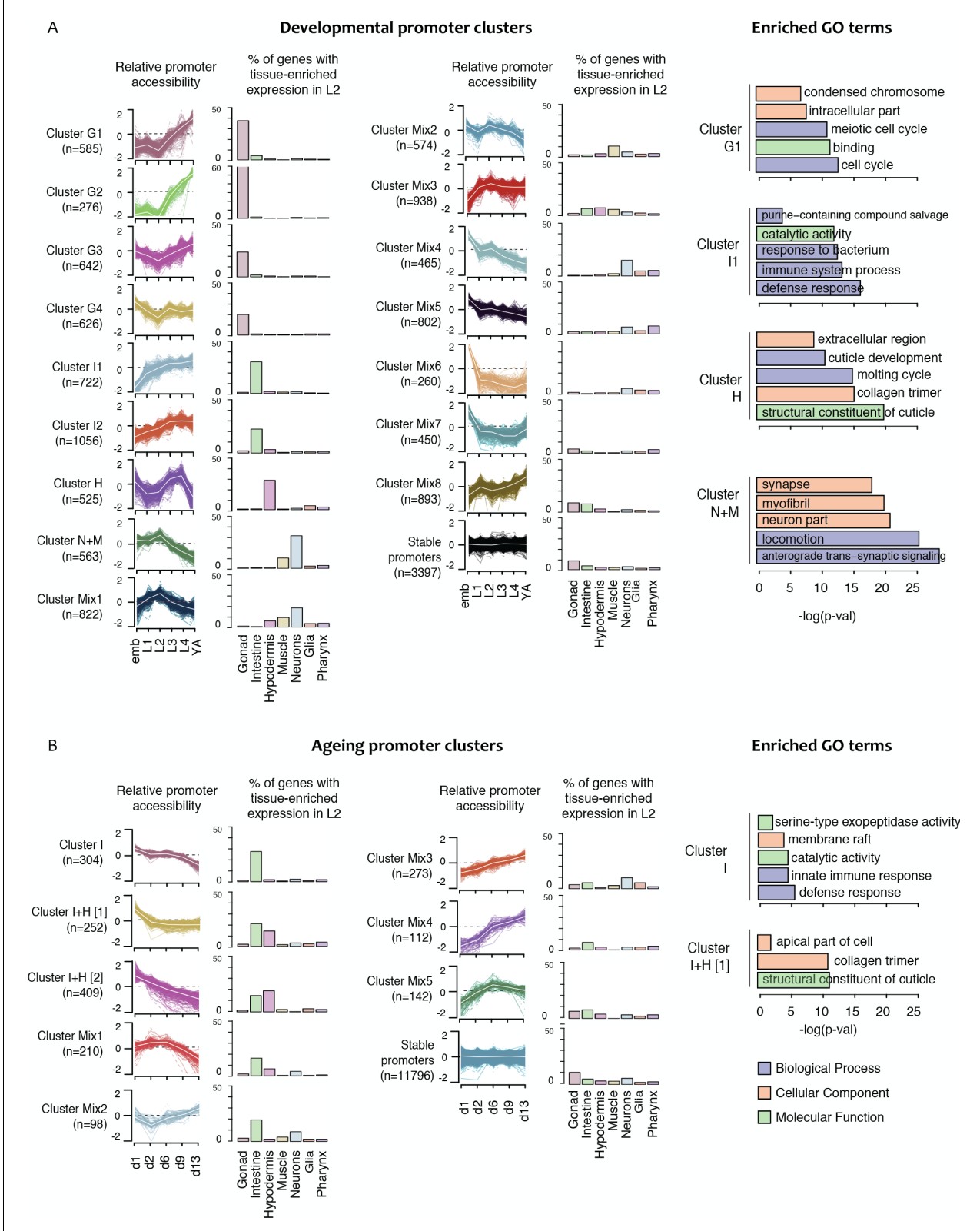

**Figure 4.** Shared dynamics of promoter accessibility in development and ageing. Clusters of promoters with shared relative accessibility patterns across (**A**) development or (**B**) ageing. Relative promoter accessibility is log2 of the depth-normalized ATAC-seq coverage at a given time point divided by the mean ATAC-seq coverage across the time series (see Materials and methods). The percentage of associated genes that have enriched expression in

*Figure 4 continued on next page*

*Figure 4 continued*

the indicated tissues was determined from single-cell L2 larval RNA-seq data (*Cao et al., 2017*); see Materials and methods). Right hand panels show examples of GO terms enriched in genes associated with development or ageing clusters.

DOI: https://doi.org/10.7554/eLife.37344.017

The following source data and figure supplements are available for figure 4:

**Source data 1.** Element accessibility dynamics and promoter accessibility clusters in development and ageing.

DOI: https://doi.org/10.7554/eLife.37344.021

**Figure supplement 1.** Characteristics of developmental promoter clusters (continued in *Figure 4—figure supplement 2*).

DOI: https://doi.org/10.7554/eLife.37344.018

**Figure supplement 2.** Characteristics of developmental promoter clusters (continued from *Figure 4—figure supplement 1*).

DOI: https://doi.org/10.7554/eLife.37344.019

**Figure supplement 3.** Characteristics of ageing promoter clusters.

DOI: https://doi.org/10.7554/eLife.37344.020

members of the DREAM complex, which controls cell cycle progression (*Figure 5A*). Taken together, the results suggest that promoters with shared accessibility patterns have shared cell- and process-specific activity, and they highlight potential regulators that are candidates for future studies.

## Analysis of ageing clusters

We next focused on chromatin accessibility changes during ageing. In contrast to the development time course, the accessibility of most promoters is stable during ageing, with only 13% (n = 1,800) of promoters showing changes (*Figure 4—source data 1*). Interestingly, 75% of these also had regulated accessibility in development.

As for the development time course, we clustered accessibility changes in ageing. We identified eight clusters of promoters with similar accessibility changes across ageing and annotated them based on tissue biases in gene expression (*Figure 4B*; *Figure 4—source data 1*). This defined one intestinal cluster (I), two clusters enriched for intestine or hypodermal biased expression (I + H) and five mixed clusters. Several mixed clusters show weak gene expression enrichments, such as intestine expression in Mix1-2 and neural expression in Mix3 (*Figure 4B*). As observed for the development clusters, enriched GO terms were consistent with gene expression biases (*Figure 4B*, *Figure 4—figure supplement 3*).

We then evaluated the enrichment of transcription factors at each ageing promoter cluster. The binding of DAF-16/FoxO, a master regulator of ageing (*Lin et al., 2001*), is associated with five ageing promoter clusters (*Figure 5B*). Consistent with a prominent role in the intestine (*Figure 4B*; *Kaplan and Baugh, 2016*), promoter clusters enriched for DAF-16 binding are also enriched for intestinal genes (*Figure 4B*). The binding enrichment patterns of five other TFs implicated in ageing (DVE-1, NHR-80, ELT-2, FOS-1 and PQM-1 (*Uno et al., 2013*; *Folick et al., 2015*; *Goudeau et al., 2011*; *Mann et al., 2016*; *Tian et al., 2016*; *Mao et al., 2016*; *Tepper et al., 2013*) are similar to DAF-16 (*Figure 5B*). These TFs and DAF-16 are also enriched in developmental intestine promoter clusters (*Figure 5A*), supporting cooperation between them in development and ageing. A group of hypodermal TFs including BLMP-1, ELT-1 and ELT-3 are found enriched at promoters in one of the two I + H ageing clusters (*Figure 5B*). Finally, CEBP-1 binding is enriched in clusters Mix3 and Mix4, which are characterized by a continuous increase of promoter accessibility across ageing. This suggests a potential role of CEBP-1 in activating a subset of genes during ageing, as it is the case for its homologue CEBP-β in mouse (*Sandhir and Berman, 2010*).

## Conclusion

For the first time, we systematically map regulatory elements across the lifespan of an animal. We identified 42,245 accessible sites in *C. elegans* chromatin and functionally annotated them based on transcription patterns at the accessible site. This avoided the problems of histone-mark-based approaches for defining element function (*Core et al., 2014*; *Henriques et al., 2018*; *Rennie et al., 2018*). Our map identified promoters active across development and ageing, but we did not find promoters for every gene. Classes that would have been missed are those for genes expressed only in males or dauer larvae (which we did not profile) and genes not active under laboratory conditions. In addition, whole-animal profiling would miss promoters active in only a small number of cells. In

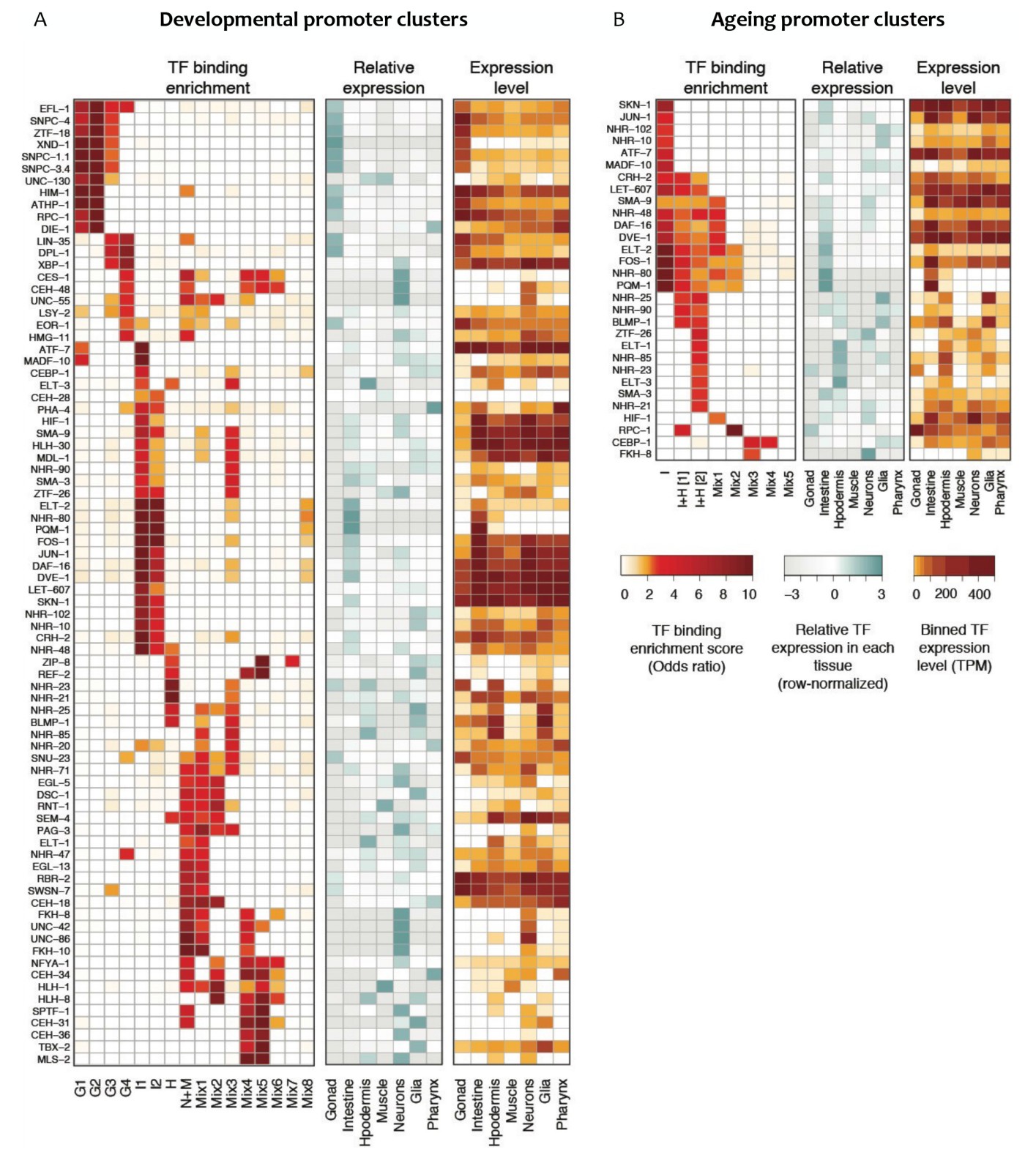

**Figure 5.** Transcription factor binding enrichment in developmental and ageing promoter clusters. Transcription factor (TF) binding enrichments in developmental (**A**) or ageing (**B**) promoter clusters from *Figure 4*. TF-binding data are from modENCODE/modERN (*Araya et al., 2014*; *Kudron et al., 2018*); peaks in HOT regions were excluded (see Materials and methods). Only TFs enriched more than twofold in at least one cluster are shown, and only enrichments with a p<0.01 (Fisher's exact test) are shown. Plots show TF binding enrichment odds ratio (left), expression of the TF

*Figure 5 continued on next page*

*Figure 5 continued*

in each tissue relative to its expression across all tissues (log2(TF tissue TPM/mean of the TF's TPMs across all tissues), middle), and the decile of expression of the TF in each tissue (right; TPMs < 1 are not taken into account when calculating TPMs deciles). Expression data are from *Cao et al. (2017)*. Legends for Figure Supplements.

DOI: https://doi.org/10.7554/eLife.37344.022

The following source data is available for figure 5:

**Source data 1.** TF datasets used for analyses.

DOI: https://doi.org/10.7554/eLife.37344.023

the future, assaying accessible chromatin and nuclear transcription in specific cell types should identify many of these missed elements.

We found that accessibility of most elements changes during the life of the worm, supporting a key role played by chromatin structure. Despite the map being based on bulk profiling in whole animals, we find that regulatory elements with shared accessibility dynamics often share patterns of tissue-specific expression, GO annotation, and TF binding. The promoters with shared accessibility changes are therefore excellent starting points for studies of cell- and process-specific gene expression. In summary, our identification of regulatory elements across *C. elegans* life together with an initial characterization of their properties provides a key resource that will enable future studies of transcriptional regulation in development and ageing.

## Materials and methods

### Collection of developmental time series samples

Wild-type N2 were grown at 20°C in liquid culture to the adult stage using standard S-basal medium with HB101 bacteria, animals bleached to obtain embryos, and the embryos hatched without food in M9 buffer for 24 hr at 20°C to obtain synchronized starved L1 larvae. L1 larvae were grown in a further liquid culture at 20°C to the desired stage, then collected, washed in M9, floated on sucrose, washed again in M9, then frozen into 'popcorn' by dripping embryo or worm slurry into liquid nitrogen. Popcorn were stored at −80°C until use. Times of growth were L1 (4 hr), L2 (20 hr), L3 (30 hr), L4 (45 hr), young adults (60 hr). Mixed populations of embryos were collected by bleaching cultures of synchronized 1-day-old adults.

### Collection of ageing time series samples

*glp-1(e2144)* were raised at 15°C on standard NGM plates seeded with OP50 bacteria. Embryos were obtained by bleaching gravid adults and then approximately 6000 placed at 25°C on 150 mm 2% NGM plates seeded with a 30X concentrated overnight culture of OP50. For harvest, worms were washed 3X in M9 and then worm slurry was frozen into popcorn by dripping into liquid nitrogen and stored at −80°C. Harvest times after embryo plating were D1/YA (53 hr), D2 (71 hr), D6 (167 hr), D9 (239 hr), D13 (335 hr).

### Nuclear isolation and ATAC-seq

Frozen embryos or worms (1–3 frozen popcorns) were broken by grinding in a mortar and pestle or smashing using a Biopulverizer, then the frozen powder was thawed in 10 ml Egg buffer (25 mM HEPES pH 7.3, 118 mM NaCl, 48 mM KCl, 2 mM CaCl2, 2 mM MgCl2). Ground worms were pelleted by spinning at 1500 g for 2 min, then resuspended in 10 ml working Buffer A (0.3M sucrose, 10 mM Tris pH 7.5, 10 mM MgCl2, 1 mM DTT, 0.5 mM spermidine 0.15 mM spermine, protease inhibitors (Roche complete, EDTA free) containing 0.025% IGEPAL CA-630. The sample was dounced 10X in a 14-ml stainless steel tissue grinder (VWR), then the sample spun 100 g for 6 min to pellet large fragments. The supernatant was kept and the pellet resuspended in a further 10 ml Buffer A, then dounced for 25 strokes. This was spun 100 g for 6 min to pellet debris and the supernatants, which contain the nuclei, were pooled, spun again at 100 g for 6 min to pellet debris, and transferred to a new tube. Nuclei were counted using a hemocytometer. One million nuclei were transferred to a 1.5-ml tube and spun 2000 g for 10 min to pellet. ATAC-seq was performed essentially as in *Buenrostro et al. (2013)*. The supernatant was removed, the nuclei resuspended in 47.5 μl of

tagmentation buffer, incubated for 30 min at 37°C with 2.5 µl Tn5 enzyme (Illumina Nextera kit), and then tagmented DNA purified using a MinElute column (Qiagen) and converted into a library using the Nextera kit protocol. Typically, libraries were amplified using 12–16 PCR cycles. ATAC-seq was performed on two biological replicates for each developmental stage and each ageing time point.

## DNAse I and MNase mapping

Replicate concentration courses of DNase I were performed for each stage as follows. Twenty million nuclei were digested in Roche DNAse I buffer for 10 min at 25°C using 2.5, 5, 10, 25, 50, 100, 200, and 800 units/ml DNase I (Roche), then EDTA was added to stop the reactions. Embryo micrococcal nuclease (MNase) digestion concentration courses for embryos were made by digesting nuclei with 0.025, 0.05, 0.1, 0.25, 0.5, 1, 4, 8, or 16 units/ml MNase in 10 mM Tris pH 7.5, 10 mM MgCl2, 4 mM CaCl2 for 10 min at 37C. Reactions were stopped by the additon of EDTA. Following digestions, total DNA was isolated from the nuclei following proteinase K and RNase A digestion, then large fragments removed by binding to Agencourt AMPure XP beads (0.5 volumes). Small double cut fragments < 300 bp were isolated either using a Pippen prep gel (protocol 1) or using Agencourt AMPure XP beads (protocol 2). Libraries were prepared as described in the Sequencing library preparation section below.

## Transcription initiation and nuclear RNA profiling

Nuclei were isolated and then chromatin associated RNA (development series) or nuclear RNA (ageing series) was isolated. Chromatin associated RNA was isolated as in (*Pandya-Jones and Black, 2009*), resuspending washed nuclei in Trizol for RNA extraction. To isolate nuclear RNA, nuclei were directly mixed with Trizol. Following purification, RNA was separated into fractions of 17–200nt and >200 nt using Zymo clean and concentrate columns. To profile transcription elongation ('long cap RNA-seq') in the nucleus, stranded libraries were prepared from the >200 nt RNA fraction using the NEB Next Ultra Directional RNA Library Prep Kit (#E7420S). Libraries were made from two biological replicates for each developmental stage and each ageing time point. To profile transcription initiation ('short cap RNA-seq'), stranded libraries were prepared from the 17–200nt RNA fraction. Non-capped RNA was degraded by first converting uncapped RNAs into 5'-monophosphorylated RNAs using RNA polyphosphatase (Epibio), then treating with 5' Terminator nuclease (Epibio). The RNA was treated with calf intestinal phosphatase to remove 5' phosphates from undegraded RNA, and decapped using Tobacco Acid Pyrophosphatase (Epicentre), Cap-Clip Acid Pyrophosphatase (CellScript, for one L2 and one L3 replicate) or Decapping Pyrophosphohydrolase (Dpph tebu-bio, for one L3 replicate) and then converted into sequencing libraries using the Illumina TruSeq Small RNA Preparation Kit kit. Libraries were size selected to be 145–225 bp long on a 6% acrylamide gel, giving inserts of 20–100 bp long. Libraries were made from two biological replicates for each developmental stage. During the course of this work, the TAP enzyme stopped being available; the Cap-Clip and Dpph enzymes perform less well than TAP. One L3 and one YA replicate was made using a slightly different protocol. Embryo short cap RNA-seq data from *Chen et al. (2013)* was also included in the analyses (GSE42819).

## ChIP-seq

Balls of frozen embryos or worms were ground to a powder using a mortar and pestle or a Retch Mixer Mill to break animals into pieces. Frozen powder was thawed into 1% formaldehyde in PBS, incubated 10 min, then quenched with 0.125M glycine. Fixed tissue was washed 2X with PBS with protease inhibitors (Roche EDTA-free protease inhibitor cocktail tablets 05056489001), once in FA buffer (50 mM Hepes pH7.5, 1 mM EDTA, 1% TritonX-100, 0.1% sodium deoxycholate, and 150 mM NaCl) with protease inhibitors (FA+), then resuspended in 1 ml FA +buffer per 1 ml of ground worm powder and the extract sonicated to an average size of 200 base pairs with a Diagenode Bioruptor or Bioruptor Pico for 25 pulses of 30 s followed by 30 s pause. For ChIP, 500 ug protein extract was incubated 2 ug antibody in FA +buffer with protease inhibitors overnight at 4°C, then incubated with magnetic beads conjugated to secondary antibodies for 2 hr at 4°C. Magnetic beads bound to immunoprecipitate were washed at room temperature twice in FA+, then once each in FA with 0.5M NaCl, FA with 1M NaCl, 0.25M LiCl (containing 1% NP-40, 1% sodium deoxycholate, 1 mM EDTA, 10 mM Tris pH8) and finally twice with TE pH8. Immunoprecipitated DNA was then eluted twice with

1% SDS, 250 mM NaCl, 10 mM Tris pH8, 1 mM EDTA at 65°C. Eluted DNA was treated with RNase for 1 hr at 37C and crosslinks reversed by overnight incubation at 65°C with 200 ug/ml proteinase K, and the DNA purified using a Qiagen column. Libraries were prepared as described in the Sequencing library preparation section below. Two biological replicate ChIPs were conducted for each histone modification at each developmental time point (Embryo, L1, L2, L3, L4, YA). Antibodies used were: anti-H3K4me3 (Abcam ab8580), anti-H3K4me1 (Abcam ab8895), anti-H3K36me3 (Abcam ab9050), and anti-H3K27me3 (Wako 309–95259).

## Sequencing library preparation

DNA was converted into sequencing libraries using a modified Illumina Truseq protocol based on https://ethanomics.files.wordpress.com/2012/09/chip_truseq.pdf. Briefly DNA fragments are first repaired with an End repair enzyme mix (New England Biolabs, cat E5060) for 30 min at 20C in 50 µl, then all DNA fragments were recovered using 1 vol of AMPure XP beads and 1 vol of 30% PEG$_{8000}$ in 1.25M NaCl, and eluted in 16.5 µl of H$_2$O. The DNA was 3' A-tailed in 1X NEB buffer 2 using 2.5 units of Klenow 3' to 5' exo(minus) (New England Biolabs, cat M0212) and 0.2 mM ATP for 30 min at 37C in 20 µl. Illumina Truseq adaptors were then directly ligated to the DNA fragments by adding 25 µl 2X buffer, 1 µl of 0.06 nM adaptors (1 µl of 1:250 dilution of Illumina stock solution), 2.5 µl water and 1.5 µl of NEB Quick ligase (cat M2200). After 20 min at room temperature, 5 µl of 0.5M EDTA pH8 was added to inactivate the enzyme and DNA was purified using AMPure XP beads. For DNAse and MNase libraries, 1.3 volumes of beads were used; for ChIP libraries, 0.9 volumes of beads were used. DNA fragments were eluted in 20 µl of H$_2$O. We used 1 µl to determine the number of cycles needed to get amplification to 50% of the plateau as in https://ethanomics.wordpress.com/ngs-pcr-cycle-quantitation-protocol/. Libraries were amplified by PCR by adding 20 µl of the KAPA Hifi Hotstart Ready Mix (Kapabiosystem cat KK2601) and 1 µl of 25 uM Illumina Universal primers. Libraries were then size selected. DNAse and MNase libraries were purified using 1.3 volumes of beads. For ChIP libraries, 0.7 volumes of beads were added to bind large DNA. Beads were discarded and DNA recovered from the supernatant by adding 0.75 volumes of beads and 0.75 volumes of 30% PEG$_{8000}$ in 1.25M NaCl. DNA was eluted in 40 µl water and 0.8 volumes of beads used to bind the library, leaving adaptor dimers in the supernatant. DNA was eluted in 10–15 µl water, quantified using a Qubit, and analyzed using a Agilent Tapestation.

## Data processing

Reads were aligned using bwa-backtrack (*Li and Durbin, 2009*) in single-end (ATAC-seq, short cap RNA-seq, ChIP-seq) or paired-end mode (ATAC-seq - developmental only, DNase-seq, MNase-seq, long cap RNA-seq). Low-quality (q < 10), mitochondrial and modENCODE-blacklisted (*Boyle et al., 2014*) reads were discarded at this point.

For ATAC-seq, normalized genome-wide accessibility profiles from single-end reads were then calculated with MACS2 (*Zhang et al., 2008*) using the parameters –format BAM –bdg –SPMR –gsize ce –nolambda –nomodel –extsize 150 –shift −75 –keep-dup all. Developmental ATAC-seq was also processed in paired-end mode (ATAC-seq libraries of ageing samples were single-end). We did not observe major differences between accessible sites identified from paired-end, and single-end profiles, and therefore use single-end profiles throughout the study for consistency.

Short-cap and long-cap data were processed essentially as in *Chen et al. (2013)*. Following alignment, and filtering, transcription initiation was represented using strand-specific coverage of 5' ends of short-cap reads. Transcription elongation was represented as strand-specific coverage of long-cap reads, with regions between read pairs filled in. For browsing, transcription elongation signal was normalized between samples by sizeFactors calculated from gene-level read counts using DESeq2 (*Love et al., 2014*). Normalized (linear) coverage signal was then further log-transformed with $log_2\left(normalised\_coverage + 1\right)$.

ChIP-seq data was processed as in *Chen et al. (2014)*. After alignment and filtering, the BEADS algorithm was used to generate normalized ChIP-seq coverage tracks (*Cheung et al., 2011*).

Stage-specific tracks used in downstream analyses were obtained by averaging normalized signal across two biological replicates. Manipulations of genome-wide signal were performed using bedtools (*Quinlan and Hall, 2010*), UCSC utilities (*Kent et al., 2010*), and wiggleTools (*Zerbino et al., 2014*). Computationally intensive steps were managed and parallelized using snakemake

(*Köster and Rahmann, 2012*). Genome-wide data was visualized using the Integrative Genomics Viewer (*Robinson et al., 2011*; *Thorvaldsdóttir et al., 2013*).

To assess the reproducibility of replicate datasets, we performed PCA using the plotPCA() function in DESeq2 (*Love et al., 2014*) on peak accessibility at promoters (ATAC-seq), read counts at annotated genes (long cap RNA-seq), 5' end read counts at promoters (short cap RNA-seq), and genic regions, from the most upstream promoter to the annotated 3' end, excluding genes with no annotated promoter (histone modifications). Replicates agreed well as shown in *Figure 1—figure supplements 2* and *3*.

## Identification of accessible sites

Accessible sites were identified as follows. We first identified concave regions (regions with negative smoothed second derivative) from ATAC-seq coverage averaged across all stages and replicates. This approach is extremely sensitive, identifying a large number (>200,000) of peak-like regions. We then scored all peaks in each sample using the magnitude of the sample-specific smoothed second derivative. We used IDR (*Li et al., 2011*) on the scores to assess stage-specific signal levels and biological reproducibility, setting a conservative cutoff at 0.001. Final peaks boundaries were set to peak accessibility extended by 75 bp on both sides. We found that calling peaks using paired end or single end data were highly similar, but some regions were captured better by one or the other. Developmental ATAC-seq datasets were sequenced paired-end and ageing datasets single-end. Peaks were therefore called separately using developmental paired-end data, developmental single-end data extended to 150 bp and shifted 75 bp upstream, and ageing (single-end only) data, and then merged. This was achieved by successively including peaks from the three sets if they did not overlap a peak already identified in an earlier set. *Figure 1—source data 1* gives peak calls and ATAC peak heights at each stage.

## Datasets and genome versions

Throughout this study, we used the WBcel215/ce10 (WS220) version of the *C. elegans* genome, and WormBase WS260 genome annotations - with coordinates backlifted to WBcel215/ce10 (WS220). For convenience, *Figure 2—source data 1* also contains WBcel235/ce11 coordinates of accessible sites and representative transcription initiation modes.

For motif analyses, Inr and TATA consensus sequences were obtained from *Sloutskin et al. (2015)*, and mapped with zero mismatches using homer (*Heinz et al., 2010*). CpG density was defined as in *Chen et al. (2014)*.

modENCODE (*Araya et al., 2014*) and modERN (*Kudron et al., 2018*) transcription factor binding datasets used in this paper were obtained from http://www.encodeproject.org or http://data.modencode.org (EOR-1). ChIP-seq profiles were manually inspected and 227 high quality datasets selected, covering 176 transcription factors (given in *Figure 5—source data 1*). To define TFBS clusters (*Figure 1—figure supplement 1C,D*; *Figure 2—figure supplement 1*), TF peak calls were extended to 200 bp on either side of the summit, and clustered using a single-linkage approach. To analyze enrichment of individual factors (*Figure 5*), TF peaks were assigned to a regulatory element if their summits overlapped with the 400 bp region centered at the element midpoint. Factors associated with each regulatory element via this approach are given in *Figure 4—source data 1*. We excluded binding at so-called 'HOT' (highly occupied target) regions from enrichment analyses in *Figure 5*, as these are thought to represent non-sequence-specific TF binding or ChIP artifacts (*Gerstein et al., 2010*; *Kudron et al., 2018*). HOT regions were defined here as accessible sites with binding of 19 or more of the analyzed 176 TFs (sites in the top 20% of binding, excluding sites with no binding).

Coefficients of variation of gene expression (CV) are from (*Gerstein et al., 2014*); processed table was kindly provided by Burak Alver).

## Annotation of regulatory elements

Patterns of nuclear transcription were used to annotate elements. At each stage, separately on both strands, we assessed 1) initiating and elongating transcription at the site, 2) continuity of transcription from the site to the closest downstream gene, and 3) positioning of nearby exons (on the matching strand).

To assess for transcription elongation at an accessible site, we counted 5′ ends of long cap reads upstream (−250:−75), and downstream (+75:+250) of peak accessibility. We then used two approaches to identify sites with a local increase in transcription elongation. First, we used DESeq2 to test for an increase in downstream vs upstream counts ('jump' method). Statistical significance was called at log2FoldChange > 1.5, and adjusted p-value<0.1 (one-sided test). To capture additional regions with weak signal ('incr' method), we accepted sites with 0 reads upstream, at least one read in both biological replicates downstream, and three reads total when summed across both biological replicates.

To assess transcription initiation, we pooled short cap across all six wild-type stages, and included two additional embryo replicates from *Chen et al. (2013)*. The pooled signal was filtered for reproducibility by only keeping signal at base pairs with non-zero transcription initiation in at least two replicates. We then required the presence of at least one base pair with reproducible signal within 125 bp of peak accessibility to designate an accessible site as having transcription initiation. For every site, we also defined a representative transcription initiation mode as the position with maximum short-cap signal within 125 bp of peak accessibility. For sites without reproducible short-cap signal, we used an extrapolated, 'best-guess' position at 60 bp downstream of peak accessibility.

We annotated accessible sites as coding_promoter or pseudogene_promoter if they fulfilled the following four criteria. (1) The accessible site had transcription initiation, and passed at least one of the elongation tests (jump or incr), or passed both elongation tests (jump and incr). (2) Transcription initiation mode at the accessible site was either upstream of the closest first exon, or, in the presence of a UTR, up to 250 bp downstream within the UTR. (The closest first exon was chosen based on the distance between the 5′ end of the first exon and peak accessibility at the accessible site, allowing the 5′ end of the exon to be up to 250 bp upstream or anywhere downstream of peak accessibility). (3) The region from peak accessibility to the closest first exon did not contain the 5′ end of a non-first exon. (4) Distal sites (peak accessibility >250 bp from the closest first exon) were additionally required to (a) have continuous long-cap coverage from 250 bp downstream of peak accessibility to the closest first exon, and (b) be further than 250 bp away from any non-first exon.

We then further attempted to assign a single, lower-confidence promoter to genes that were not assigned a promoter so far. For every gene without promoter assignments, we re-examined sites that fulfilled criteria (2-4), and were either intergenic, or within 250 bp of the closest first exon. We then annotated the site with the largest jump test log2FoldChange as the promoter, if it was also larger than 1.

Next, sites within 250 bp of the 5′ end of an annotated tRNA, snRNA, snoRNA, miRNA, or rRNA were annotated as non-coding_RNA. Intergenic sites more than 250 bp away from annotated exons that had initiating transcription, and passed the jump test were annotated as unassigned_promoter. All remaining sites were annotated as transcription_initiation or no_transcription based on whether they had transcription initiation.

Elements were then annotated on each strand based on aggregating transcription patterns across stages by determining the 'highest' annotation using the ranking of: coding_promoter, pseudogene_promoter, non-coding_RNA, unassigned_promoter, transcription_initiation, no_transcription. Element type and coloring was then defined using the following ranking: coding_promoter on either strand => coding_promoter (red); pseudogene_promoter on either strand => pseudogene_promoter (orange); non-coding_RNA on either strand => non-coding_RNA (black); unassigned_promoter on either strand => unassigned_promoter (yellow); transcription_initiation on either strand => putative_enhancer (green); all remaining sites => other_element (blue). *Figure 2—source data 1* gives annotation information.

## Clustering of promoter accessibility

Accessible elements with regulated accessibility were determined as follows. All elements (n = 42,245) were tested for a difference in ATAC-seq coverage between any two developmental time points or between any two ageing time points using DESeq2 (*Love et al., 2014*). Sites with >= 2 absolute fold change and adjusted p-value<0.01 were defined as 'regulated' (n = 30,032 in development and n = 6590 in ageing; *Figure 4—source data 1*); regulated promoters (n = 10,199 in development and n = 1800 in ageing) were used in clustering analyses.

For clustering analyses, depth-normalized ATAC-seq coverage of each promoter was calculated at each time point in development or ageing. Relative accessibility was calculated at each time point

in development or ageing by applying the following formula: $log_2 \left( ATACseq\ coverage\ _{time\ point\ i} + 1 \right) - log_2(mean\ ATACseq\ coverage\ across\ time\ points + 1)$. Mean ATAC-seq coverage across time points was calculated separately for the developmental and ageing time courses. Clustering was performed using *k*-medoids, as implemented in the pam() method of the cluster R package (*Maechler et al., 2017*). Different numbers of clusters were tested for clustering of regulatory elements in developmental and ageing datasets; 16 was chosen for developmental data and 10 for ageing data as the normalized changes in promoter ATAC-seq signals within each cluster were relatively homogeneous. We manually merged two ageing clusters showing comparable accessibility and tissue-specific gene enrichment (resulting in the cluster I + H [2]). Clusters labels were determined based on enrichment for tissue-biased gene expression within each cluster (see below).

To compare accessibility and gene expression, FPM-normalized gene-level read counts were calculated using DESeq2, and then averaged across biological replicates. For visualisation, relative expression levels were calculated using the approach described above for relative promoter accessibility (see formula above), with FPM values instead of ATAC-seq coverage values.

Using single-cell RNA-seq data from *Cao et al. (2017)*, we defined tissue-biased gene expression as follows: Gene expression was considered enriched in a given tissue if it had a fold-change >= 3 between expression in the tissues with highest and second highest levels and an adjusted p-value<0.01. This defined 5315 genes with tissue-biased expression (1432 in Gonad, 553 in Hypodermis, 799 in Intestine, 352 in Muscle, 1218 in Neurons, 447 enriched in Glia, 514 in Pharynx). For each developmental or ageing cluster of promoters, we calculated the percentage of genes with biased expression in a given tissue relative to the total number of genes in the cluster. These values were plotted in *Figure 4A and B* (bar plots).

GO enrichments were evaluated using the R package gProfileR (*Reimand et al., 2016*) against *C. elegans* GO database. Significant enrichment was set at an adjusted p-value of 0.05, and hierarchically redundant terms were automatically removed by gProfileR.

## Enrichment for transcription factor binding in promoter clusters

Prior to analysis of TF peak enrichment at annotated promoters, accessible elements considered 'HOT' (see above) were removed, resulting in 10,086 to be assessed by enrichment analysis. Only transcription factors with more than 200 peaks overlapping 'non-hot' regulatory elements were kept, to ensure sufficient data for analysis. Following this stringent filtering, 89 transcription factors could be assayed for binding enrichment. Transcription factor binding enrichment in each cluster was estimated using the odds ratio and enrichments with an associated p-value<0.01 (Fisher's exact test) were kept. Transcription factors which did not show enrichment higher than two in any cluster were discarded. *Figure 5* summarizes the transcription factor binding enrichment in each cluster during development or ageing. Relative tissue expression profiles of each transcription factor at the L2 stage (data from *Cao et al., 2017*) was calculated in each tissue by taking the log2 of its expression (TPM) in the tissue divided by its mean expression across all tissues. A pseudo-value of 0.1 was first added to all the TPM values before calculation of the relative levels of expression.

## Construction of transgenic lines

Transgene constructs were made using three-site Gateway cloning (Invitrogen) as in *Chen et al. (2014)*. Site one has the regulatory element sequence to be tested, site two has a synthetic outron (OU141; *Conrad et al., 1995*) fused to *his-58* (plasmid pJA357), and site three has gfp-tbb-2 3'UTR (pJA256; *Zeiser et al., 2011*) in the MosSCI compatible vector pCFJ150, which targets Mos site Mos1(ttTi5605); MosSCI lines were generated as described (*Frøkjaer-Jensen et al., 2008*).

## Data access

ATAC-seq, ChIP-seq, DNase/MNase-seq, long/short cap RNA-seq data from this study, including processed tracks are available at the NCBI Gene Expression Omnibus (GEO) (http://www.ncbi.nlm.nih.gov/geo/) under accession number GSE114494.

## Acknowledgements

We thank C Bradshaw for bioinformatics support, K Harnish for sequencing, B Alver for providing processed data, and F Carelli, C Gal, and A Frapporti for comments on the manuscript. The work was supported by Wellcome Trust Senior Research Fellowships to JA (054523 and 101863), a Wellcome Trust PhD fellowship to JJ (097679), a Sir Robert Edwards Scholarship from Churchill College, an English Speaking Union Graduate Scholarship, and funding from the Cambridge Trust to MS, a Medical Research Council DTP studentship to JS, and a Thouron award to CW. This study was also supported by the European Sequencing and Genotyping Infrastructure (funded by the EC, FP7/2007-2013) under Grant Agreement 26205 (ESGI) as part of the transnational access program. We thank Drs. Hans Lehrach and Marie-Laure Yaspo for generous support of the ESGI project, Dr. Marc Sultan for setting up sequencing technology platforms, and Mathias Linser and the rest of the sequencing team of the Department of Vertebrate Genomics at the Max Planck Institute for Molecular Genetics for technical assistance. We also acknowledge core support from the Wellcome Trust (092096) and Cancer Research UK (C6946/A14492).

## Additional information

### Competing interests

Julie Ahringer: Reviewing editor, *eLife*. The other authors declare that no competing interests exist.

### Funding

| Funder | Grant reference number | Author |
| --- | --- | --- |
| Wellcome | 101863 | Jürgen Jänes<br>Yan Dong<br>Alex Appert<br>Chiara Cerrato<br>Ron Chen<br>Carolina Gemma<br>Ni Huang<br>Przemyslaw Stempor<br>Annette Steward<br>Eva Zeiser<br>Julie Ahringer |
| Medical Research Council | | Jacques Serizay |
| European Commission | FP7/2007-2013 | Sascha Sauer<br>Julie Ahringer |
| Wellcome | 097679 | Jürgen Jänes |

The funders had no role in study design, data collection and interpretation, or the decision to submit the work for publication.

### Author contributions

Jürgen Jänes, Conceptualization, Data curation, Software, Formal analysis, Validation, Investigation, Visualization, Methodology, Writing—original draft, Writing—review and editing; Yan Dong, Investigation, Methodology, Writing—review and editing; Michael Schoof, Alex Appert, Formal analysis, Investigation, Methodology; Jacques Serizay, Software, Formal analysis, Visualization, Writing—original draft, Writing—review and editing; Chiara Cerrato, Carson Woodbury, Ron Chen, Carolina Gemma, Djem Kissiov, Annette Steward, Eva Zeiser, Formal analysis, Investigation; Ni Huang, Software, Formal analysis; Przemyslaw Stempor, Data curation, Software, Formal analysis; Sascha Sauer, Funding acquisition, Project administration; Julie Ahringer, Conceptualization, Formal analysis, Supervision, Funding acquisition, Writing—original draft, Project administration, Writing—review and editing

**Author ORCIDs**
Jürgen Jänes (iD) http://orcid.org/0000-0002-2540-1236
Ni Huang (iD) http://orcid.org/0000-0001-8849-038X
Przemyslaw Stempor (iD) http://orcid.org/0000-0002-9464-7475
Julie Ahringer (iD) http://orcid.org/0000-0002-7074-4051

**Decision letter and Author response**
Decision letter https://doi.org/10.7554/eLife.37344.038
Author response https://doi.org/10.7554/eLife.37344.039

## Additional files

**Supplementary files**
• Transparent reporting form
DOI: https://doi.org/10.7554/eLife.37344.024

**Data availability**
Sequencing data have been deposited in as a SuperSeries in GEO under accession code GSE114494.

The following datasets were generated:

| Author(s) | Year | Dataset title | Dataset URL | Database and Identifier |
|---|---|---|---|---|
| Julie Ahringer, Jürgen Jänes | 2018 | Chromatin accessibility dynamics across C. elegans development and ageing [DNase, MNase] | https://www.ncbi.nlm.nih.gov/geo/query/acc.cgi?acc=GSE114481 | Gene Expression Omnibus, GSE114481 |
| Ahringer J, Jürgen Jänes | 2018 | Chromatin accessibility dynamics across C. elegans development and ageing [scap] | https://www.ncbi.nlm.nih.gov/geo/query/acc.cgi?acc=GSE114490 | NCBI Gene Expression Omnibus, GSE114490 |
| Julie Ahringer, Jürgen Jänes | 2018 | Chromatin accessibility dynamics across C. elegans development and ageing [lcap] | https://www.ncbi.nlm.nih.gov/geo/query/acc.cgi?acc=GSE114483 | Gene Expression Omnibus, GSE114483 |
| Julie Ahringer, Jürgen Jänes | 2018 | Chromatin accessibility dynamics across C. elegans development and ageing [ChIP-seq] | https://www.ncbi.nlm.nih.gov/geo/query/acc.cgi?acc=GSE114440 | Gene Expression Omnibus, GSE114440 |
| Julie Ahringer, Jürgen Jänes | 2018 | Chromatin accessibility dynamics across C. elegans development and ageing [ATAC-seq] | https://www.ncbi.nlm.nih.gov/geo/query/acc.cgi?acc=GSE114439 | Gene Expression Omnibus, GSE114439 |

The following previously published datasets were used:

| Author(s) | Year | Dataset title | Dataset URL | Database and Identifier |
|---|---|---|---|---|
| Down TA | 2013 | The landscape of RNA polymerase II transcription initiation in C. elegans reveals a novel regulatory architecture | https://www.ncbi.nlm.nih.gov/geo/query/acc.cgi?acc=GSE42819 | NCBI Gene Expression Omnibus, GSE42819 |

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
