## [Decision Letter]

Thank you for submitting your article "Chromatin accessibility dynamics across *C. elegans* development and ageing" for consideration by *eLife*. Your article has been reviewed Jessica Tyler as the Senior Editor, a Reviewing Editor, and three reviewers. The following individual involved in review of your submission has agreed to reveal her identity: Bérénice A Benayoun (Reviewer #1).

The reviewers have discussed the reviews with one another and the Reviewing Editor has drafted this decision to help you prepare a revised submission.

Summary:

The reviewers appreciated the high impact of the work, particularly as an important resource for the community. The reviewers generally agreed that the experimental designs and data analyses were rigorous and the results were well described.

Essential revisions:

1) A major point that needs to be addressed is the potential caveats relate to changes in cell numbers as pointed out by reviewer 2. This can be addressed textually, however the corresponding conclusions will need to modified accordingly.

2) The reviewers suggested a number of clarifications, including additional quality control analyses, and clear discussion of differences in methods. These include all specific comments from reviewers 1 and 3 below.

Reviewer #1:

This manuscript by Janes et al., described a novel dataset resource mapping chromatin elements throughout *C. elegans* lifespan, from development to adulthood, and, for the first time in worm, throughout several stages of adulthood. The authors identify > 40K elements with accessible chromatin in at least one of the assayed time points, with highly dynamic landscape of accessibility throughout worm lifespan. Using nuclear transcription profiling, to bypass the *C. elegans* specific trans-splicing phenomenon, the authors refine annotations of >15K promoter and > 19K enhancers. Both classes of elements seem to be able to drive bidirectional transcription based on follow-up reporter experiments in 2 lines.

The article does describe the resource dataset well, and the raw data is already readily available. This resource will be of broad interest to the aging and chromatin fields. A couple of points need to be further clarified, and some complementary analyses need to be performed for reproducibility and statistical soundness.

1) The young adult stage profiling is performed in 2 distinct contexts: WT N2 and the *glp-1* mutant, serving as an anchor point between the developmental and adult datasets. The rationale for the switch is reasonable, but it would be important to discuss how similar/dissimilar the YA ATAC-seq data is between the two strains (N2 and *glp-1* mutant).

2) In the Discussion section of similarity vs. dissimilarity with previous developmental datasets, it would also be important to include the state of culture: liquid vs. solid, as this may have broad consequences on gene activation patterns, notably in the muscle tissue. The Daugherty et al., paper used solid cultures, whereas this study had liquid cultures (according to subsection “Collection of developmental time series samples”). It would be important to include a significance of correlation test.

3) In the CV analysis (subsection “Patterns of histone marks at promoters and enhancers”/Figure 3), it may be useful to include a statistical analysis, for instance in the form of a significance of correlation test, and maybe a scatter plot.

4) In the Materials and methods section, the authors mention the use of "the Illumina TruSeq kit or a homemade equivalent". For the sake of reproducibility, it is important to include (i) a table listing which dataset was generated which each kit/method and (ii) a description of the "homemade equivalent (steps, enzymes, used suppliers).

5) For the tissue-specific enrichment analysis (Materials and methods section), the use of a background and how it was selected needs to be included. This choice can greatly affect observed enrichment results.

6) Metagene analysis are convenient representation tools but (i) they smooth away variations in the data, potentially masking heterogeneity, and (ii) have no statistical support for changes described. To fully support conclusions, the authors need to include a complementary statistical analysis for the data reported in Figure 3B, Figure 1—figure supplement 1C/D, Figure 2—figure supplement 1B/C, Figure 3—figure supplement 1B. This can be done by either including the 95% confidence interval on the metagene plots or performing a quantitative analysis using a boxplot and a non-parametric Wilcoxon rank-sum test.

*Reviewer #2:*

The authors present whole animal ATAC-seq data for six developmental stages of *C. elegans* along with five stages of aging adults to define accessible sites. They also collect long nuclear RNA-seq data for each time point in both time series to help classify the accessible sites into promoter and enhancer types. To further augment the developmental series in particular, the authors collected chromatin ChIP-seq data and short, capped nuclear RNA to assess chromatin state and transcription initiation, respectively. After assigning a type to the accessible regions, they cluster the promoter sites based on their accessibility over the developmental time course and the aging time course. Using single cell data from Cao et al., (2017) they find that some site clusters are associated with target genes with tissue-specific expression. They also use the single cell data in conjunction with ChIP-seq TF data to show that some TFs are preferentially associated with genes expressed in certain tissues.

Overall, the paper is well-written and the data sets seem of high quality. The chromatin accessibility data should be of considerable interest to the *C. elegans* community, and their evaluation of chromatin marks relative to promoters, enhancers, and the patterns of expression of the target genes should be of interest to the wider chromatin community. The fact that most sites change over time is not surprising, given that the expression of most genes changes considerably over time. This conclusion and the cluster analysis is complicated by the fact that the fraction of cells from any given tissue changes over time, and thus the fraction of reads from a peak associated with that tissue will change, even if accessibility at that site within that cell does not change over time. The clearest example is the gonad, where cells from this tissue go from being less than 1% of the total cell (genome) number to more than half over the course of larval development. An accessible site in gonad cells that was constant throughout this expansion would be expected to show a large increase over time, simply from the increasing cell number. Similarly, intestinal cells double their genomes with each larval molt. Neurons and muscle increase in L1 but not thereafter, so proportionally the signal from their accessible sites would be expected to diminish over time, even if accessibility remained constant. Indeed, the patterns of most of the assigned clusters for the developmental time course follow the expectation from cell numbers. Finally, the analysis of TF ChIP-seq data seems quite similar to that of Cao et al., except that the authors condition their analysis on overlap of ChIP-seq sites with accessible sites. However, since the overlap of accessible sites with ChIP-seq sites is very high, it is not clear what is new here.

In sum, the paper presents data sets that will provide an important resource for the worm community. The biological insights are modest in the present paper, but its utility to the community should produce valuable insights over time.

*Reviewer #3:*

The Janes et al., manuscript reports ATAC seq profiling along a developmental and aging time course in whole worm *C. elegans*. In general, the experimental design and the data processing all appear rigorous and well done. By comparing the ATAC data with RNA-seq of nuclear RNAs, the authors define many promoters and putative enhancers. Moreover, the authors found that many of the accessible sites show dynamic developmental regulation, and the sites are often associated with tissue-specific gene expression. Overall, the data reported represent an important resource for the community.

1) It will be important to show quality control analyses to assess the reproducibility of the data / between replicates.

2) Similarly, MDA or PCA to show the overall relatedness of the data from the different time points will be helpful.

3) For the ATAC data comparison with previously published data, the authors indicated that different peak calling parameters likely account for the differences. It will be helpful for the authors to examine their data using the previously published analysis pipelines (or vice versa) and provide some quantification of the differences / similarities between the datasets.

---

## [Author Response]

Essential revisions:Reviewer #1:This manuscript by Janes et al., described a novel dataset resource mapping chromatin elements throughout C. elegans lifespan, from development to adulthood, and, for the first time in worm, throughout several stages of adulthood. The authors identify > 40K elements with accessible chromatin in at least one of the assayed time points, with highly dynamic landscape of accessibility throughout worm lifespan. Using nuclear transcription profiling, to bypass the C. elegans specific trans-splicing phenomenon, the authors refine annotations of >15K promoter and > 19K enhancers. Both classes of elements seem to be able to drive bidirectional transcription based on follow-up reporter experiments in 2 lines.The article does describe the resource dataset well, and the raw data is already readily available. This resource will be of broad interest to the aging and chromatin fields. A couple of points need to be further clarified, and some complementary analyses need to be performed for reproducibility and statistical soundness.1) The young adult stage profiling is performed in 2 distinct contexts: WT N2 and the glp-1 mutant, serving as an anchor point between the developmental and adult datasets. The rationale for the switch is reasonable, but it would be important to discuss how similar/dissimilar the YA ATAC-seq data is between the two strains (N2 and glp-1 mutant).

We make no comparisons between the developmental and ageing datasets in this paper, hence we do not discuss similarities and differences. The primary difference between wild-type N2 and *glp-1* young adults is the absence of germ line tissue in *glp-1* young adults. To validate this difference in the ATAC-seq data and address the reviewer’s question about similarities and differences, we compared signal at protein-coding promoters based on the tissue with maximum expression according to (Cao et al., 2017). Promoters of genes with the highest expression in somatic tissues show correlations between 0.87 and 0.90 (Spearman rank correlation), validating similarity between wild-type and glp-1 somatic tissue. As expected, promoters of genes with the highest expression in the gonad show a lower correlation of 0.71. Many of these latter promoters are ubiquitously active, and so also have ATAC-seq signal in somatic tissues, leading to a relatively high correlation.

**Author response image 1. respfig1:** Correlation of peak chromatin accessibility between wild-type young adult, and glp-1 day 1 time points at protein-coding promoters, grouped by tissue with the highest gene expression.

Spearman's rank correlation was calculated between peak accessibility at protein-coding promoters between wild-type young adult, and glp-1 day 1 time points. Promoters were grouped by the tissue with highest gene expression of the target gene, according to L2 single-cell gene-expression data from (Cao et al. 2017).</Author response image 1 title/legend>

2) In the Discussion section of similarity vs. dissimilarity with previous developmental datasets, it would also be important to include the state of culture: liquid vs. solid, as this may have broad consequences on gene activation patterns, notably in the muscle tissue. The Daugherty et al., paper used solid cultures, whereas this study had liquid cultures (according to subsection “Collection of developmental time series samples”). It would be important to include a significance of correlation test.

In the response to point 1 above, we compared accessibility of elements in wild-type young adults grown in liquid culture and *glp-1* young adults grown on solid media. Notably, ATAC-seq signals at regulatory elements of genes with muscle biased expression show equally high correlation compared to those of other somatic tissues, suggesting that media type is unlikely to play a major role in accessibility differences. Nevertheless, we added to the paper that such a difference could have some contribution. Further, in response to this point and reviewer 3 point 3, we have expanded the analyses (adding new Figure 2—figure supplement 2. Effect of differences in peak calling methods on the types of identified accessible sites) and discussion of similarities and differences between previous datasets (see below).

3) In the CV analysis (subsection “Patterns of histone marks at promoters and enhancers”/Figure 3), it may be useful to include a statistical analysis, for instance in the form of a significance of correlation test, and maybe a scatter plot.

We have added a p-value confirming the statistical significance of the correlation.

4) In the Materials and methods section, the authors mention the use of "the Illumina TruSeq kit or a homemade equivalent". For the sake of reproducibility, it is important to include (i) a table listing which dataset was generated which each kit/method and (ii) a description of the "homemade equivalent (steps, enzymes, used suppliers).

We have added a section to the Materials and methods section describing the protocol used to prepare the sequenced libraries.

5) For the tissue-specific enrichment analysis (Materials and methods section), the use of a background and how it was selected needs to be included. This choice can greatly affect observed enrichment results.

We have clarified this in the Materials and methods section as follows:

“Using single cell RNA-seq data from (Cao et al., 2017), we defined tissue-biased gene expression as follows: Gene expression was considered enriched in a given tissue if it had a fold-change >= 3 between expression in the tissues with highest and second highest levels and an adjusted p-value < 0.01. This defined 5,315 genes with tissue-biased expression (1432 in Gonad, 553 in Hypodermis, 799 in Intestine, 352 in Muscle, 1218 in Neurons, 447 enriched in Glia, 514 in Pharynx). For each developmental or ageing cluster of promoters, we calculated the percentage of genes with biased expression in a given tissue relative to the total number of genes in the cluster. These values were plotted in Figure 4 A and B (bar plots).”

6) Metagene analysis are convenient representation tools but (i) they smooth away variations in the data, potentially masking heterogeneity, and (ii) have no statistical support for changes described. To fully support conclusions, the authors need to include a complementary statistical analysis for the data reported in Figure 3B, Figure 1—figure supplement 1C/D, Figure 2—figure supplement 1B/C, Figure 3—figure supplement 1B. This can be done by either including the 95% confidence interval on the metagene plots or performing a quantitative analysis using a boxplot and a non-parametric Wilcoxon rank-sum test.

As requested, we have added 95% confidence intervals to Figure 2—figure supplement 1B,C; Figure 3B; Figure 3—figure supplement 1B. For Figure 1—figure supplement 1C and D only two data points (replicates) were available for each comparison group (assay/condition). We therefore highlighted the signal range for every group.

Reviewer #2:The authors present whole animal ATAC-seq data for six developmental stages of C. elegans along with five stages of aging adults to define accessible sites. They also collect long nuclear RNA-seq data for each time point in both time series to help classify the accessible sites into promoter and enhancer types. To further augment the developmental series in particular, the authors collected chromatin ChIP-seq data and short, capped nuclear RNA to assess chromatin state and transcription initiation, respectively. After assigning a type to the accessible regions, they cluster the promoter sites based on their accessibility over the developmental time course and the aging time course. Using single cell data from Cao et al., (2017) they find that some site clusters are associated with target genes with tissue-specific expression. They also use the single cell data in conjunction with ChIP-seq TF data to show that some TFs are preferentially associated with genes expressed in certain tissues.Overall, the paper is well-written and the data sets seem of high quality. The chromatin accessibility data should be of considerable interest to the C. elegans community, and their evaluation of chromatin marks relative to promoters, enhancers, and the patterns of expression of the target genes should be of interest to the wider chromatin community. The fact that most sites change over time is not surprising, given that the expression of most genes changes considerably over time. This conclusion and the cluster analysis is complicated by the fact that the fraction of cells from any given tissue changes over time, and thus the fraction of reads from a peak associated with that tissue will change, even if accessibility at that site within that cell does not change over time. The clearest example is the gonad, where cells from this tissue go from being less than 1% of the total cell (genome) number to more than half over the course of larval development. An accessible site in gonad cells that was constant throughout this expansion would be expected to show a large increase over time, simply from the increasing cell number. Similarly, intestinal cells double their genomes with each larval molt. Neurons and muscle increase in L1 but not thereafter, so proportionally the signal from their accessible sites would be expected to diminish over time, even if accessibility remained constant. Indeed, the patterns of most of the assigned clusters for the developmental time course follow the expectation from cell numbers.

When clustering promoter accessibility changes across development, clear patterns are visible. As noted by the reviewer, in some cases these patterns follow changes in cell number during development. However, this does not invalidate the clustering analysis, which aimed at identifying groups of promoters with similar patterns of accessibility across development or ageing. The patterns could be due to changes in accessibility or changes in cell number. For example, through this method we identified different groups of germ line active promoters that have greatly increased accessibility signals as the germ cell number increases, that are active in the germ line. Other patterns, such as oscillating accessibility (observed in Cluster H) or decreased accessibility after embryogenesis (Clusters Mix6 and Mix7) are likely to be due to accessibility regulation rather than changes in cell number. We clarified in the manuscript that the accessibility changes may sometimes reflect changes in cell numbers rather than the accessibility dynamics across development.

Reviewer #3:The Janes et al., manuscript reports ATAC seq profiling along a developmental and aging time course in whole worm C. elegans. In general, the experimental design and the data processing all appear rigorous and well done. By comparing the ATAC data with RNA-seq of nuclear RNAs, the authors define many promoters and putative enhancers. Moreover, the authors found that many of the accessible sites show dynamic developmental regulation, and the sites are often associated with tissue-specific gene expression. Overall, the data reported represent an important resource for the community.1) It will be important to show quality control analyses to assess the reproducibility of the data / between replicates.2) Similarly, MDA or PCA to show the overall relatedness of the data from the different time points will be helpful.

We have added PCA plots showing reproducibility and broad relatedness of the samples and assays in Figure 1—figure supplement 2.

3) For the ATAC data comparison with previously published data, the authors indicated that different peak calling parameters likely account for the differences. It will be helpful for the authors to examine their data using the previously published analysis pipelines (or vice versa) and provide some quantification of the differences / similarities between the datasets.

As suggested, we used MACS2 to call peaks on our ATAC-seq data as done by Daugherty et al., 2017 and then compared this set to the accessible sites presented in this paper, which were identified using a focal enrichment peak calling method. Overall, the MACS2 peak calls are very similar to the accessible sites reported in this study (91.5% overlap; Figure 2—figure supplement 2A). The overlapping sites are enriched for ChIP-seq peaks and transcription initiation signal, and they are depleted for mapping at exons. The accessible sites that do not overlap MACS2 peak calls are similar to these, showing transcription initiation signal and depletion for exons, but this set is depleted for ChIP-seq peaks suggesting they may be sites active in a small number of cells. The small fraction of MACS2 peak calls that do not overlap an accessible site (8.5%, n=2392) are similar to the peak calls found by Daugherty et al., 2017 or Ho et al., 2017 but that are not present in our accessible sites. They are depleted for ChIP-seq peaks and transcription initiation signal, and they are enriched in annotated exons. The difference in peak calling methods therefore can account for some of the differences with previously published data. However, the fraction of such sites is relatively small indicating that data specific differences also contribute. Indeed, the peaks defined using MACS2 on the ATAC-seq data reported here still show substantial differences with those reported by Daugherty et al., 2017 (Figure 2—figure supplement 2B). We include this analysis as a new figure (Figure 2—figure supplement 2B) and discuss the results in the text (subsection “Defining and annotating regions of accessible DNA”). Contributing factors may be differences in signal to noise or differences in growth methods (liquid vs solid media).